# Deep Learning Detection and Segmentation of Facet Joints in Ultrasound Images Based on Convolutional Neural Networks and Enhanced Data Annotation

**DOI:** 10.3390/diagnostics14070755

**Published:** 2024-04-02

**Authors:** Lingeer Wu, Di Xia, Jin Wang, Si Chen, Xulei Cui, Le Shen, Yuguang Huang

**Affiliations:** Department of Anesthesiology, Peking Union Medical College Hospital, Chinese Academy of Medical Sciences and Peking Union Medical College, Beijing 100006, China; 13426365656@126.com (L.W.); xiad08@163.com (D.X.); wangjin05@163.com (J.W.); chensi93@pumch.cn (S.C.); pumchshenle@aliyun.com (L.S.); garybeijing@163.com (Y.H.)

**Keywords:** ultrasound image, deep learning, facet joint, convolutional neural network, enhanced data annotation, ventral complex

## Abstract

The facet joint injection is the most common procedure used to release lower back pain. In this paper, we proposed a deep learning method for detecting and segmenting facet joints in ultrasound images based on convolutional neural networks (CNNs) and enhanced data annotation. In the enhanced data annotation, a facet joint was considered as the first target and the ventral complex as the second target to improve the capability of CNNs in recognizing the facet joint. A total of 300 cases of patients undergoing pain treatment were included. The ultrasound images were captured and labeled by two professional anesthesiologists, and then augmented to train a deep learning model based on the Mask Region-based CNN (Mask R-CNN). The performance of the deep learning model was evaluated using the average precision (AP) on the testing sets. The data augmentation and data annotation methods were found to improve the AP. The AP50 for facet joint detection and segmentation was 90.4% and 85.0%, respectively, demonstrating the satisfying performance of the deep learning model. We presented a deep learning method for facet joint detection and segmentation in ultrasound images based on enhanced data annotation and the Mask R-CNN. The feasibility and potential of deep learning techniques in facet joint ultrasound image analysis have been demonstrated.

## 1. Introduction

Pain is a common medical condition. Pain diseases ranked first among all diseases according to the Global Burden of Disease study published by the Lancet in 2018, with lower back pain being the primary cause of motion limitation, causing loss of labor ability, in most countries [1]. Up to 80% of people experience chronic neck pain and lower back pain during their lifetime [2]. Facet joint conditions are the most common causes of chronic spinal-derived lower back pain [3,4]. 

Each facet joint is located at the junction of the pedicle and the laminae, which consist of the upper facet joint and lower facet joint of the adjacent vertebrae. Facet joints are the only synovial joints in the spine, and their surfaces are covered by hyaline cartilage. The capsule contains synovial fluid. The articular surface consists of the outer fibrous capsule and the inner synovial capsule. There are non-myelinated notional receptors and myelinated mechanoreceptors distributed in each joint capsule, which means that the facet joints play an essential role in maintaining spinal stability and regular physiological activity.

In clinical practice, 15–40% of lower back pain is caused by degeneration of the lumbar facet joints. Conservative treatment for articular pain of the lumbar facet joints is mainly used, such as hot compresses, short-wave ultrasounds, and oral nonsteroidal anti-inflammatory drugs. Although lower back pain can be temporarily alleviated, the long-term effect is not so satisfactory. With the development of minimally invasive techniques, interventional therapy has become a safer and more effective method for the treatment of lower back pain caused by facet joint conditions. Facet joint injections are one of the most common procedures performed by pain management anesthesiologists [5]. Injections can be directed by fluoroscopy, computed tomography (CT), and palpation or loss-of-resistance techniques [6]. Techniques such as fluoroscopy and CT have significant disadvantages. For example, the patients will be exposed to ionizing radiation, which means that these techniques may be not appropriate for patients who are pregnant. In addition, it has been reported that the palpation and loss-of-resistance techniques for epidural injections have a failure rate of 6% to 20% [7]. Alternatively, ultrasound, a real-time nonionizing imaging technique, has the capacity to visualize soft tissues and bony surfaces, and has been increasingly utilized for facet joint injections in recent years [8,9]. For instance, Overnauer et al. [10] compared ultrasound-guided and CT-guided facet joint injections. Their results revealed that injections guided by ultrasound were faster than and presented the same therapeutic effects as CT-guided injections [10]. Additionally, Wang et al. [11] compared image guidance technologies for interventional pain procedures. Their study revealed that although ultrasound guidance is beneficial in spinal injections, the success rate of the procedure still depends greatly on the experience of the anesthesiologists [11]. It requires a long learning curve for most pain specialists to be familiar with ultrasound guidance techniques [6]. That means the visualization and identification of ultrasound imaging, especially for the facet joints, remains a problem for anesthesiologists. Artificial intelligence (AI) is an emerging technology for addressing the above issues.

AI has become one of the most popular tools for medical image analysis. The convolutional neural networks (CNNs) are among the AI techniques [12,13] which have been applied to ultrasound images to identify and recognize different target objects, such as neural vascular structures [14], left ventricles [15], breast tumors [16], and the spine [17]. CNNs may gradually become able to interpret low-level features as if they were high-level features, which are mainly applied to object detection and recognition in image and video analysis. Deep learning network models based on object detection are increasingly used in medical image processing, especially in spinal ultrasounds, including spinal image recognition, disease detection, and disease prediction. However, deep learning network models based on object detection need to be based on a large amount of data. In fact, the current medical image training data scale is significantly smaller than the public data sets used in other fields such as natural image understanding, which results in lower prediction performance. The current CNN model is mainly limited to the single region features of fixed morphology. However, the facet joints are often shown in ultrasonic images with different scales, unclear edges, and irregular shapes, etc., which cannot be thoroughly characterized by single region features.

The ventral complex, located in the ventral dural space, is a complex composed of tissues such as the ventral dural membrane and the anterior longitudinal ligament, showing a high echo zone on ultrasound. The ventral complex plays an essential role in the indication of ultrasound-guided intraspinal puncture. Its presence usually indicates that an ultrasound beam can pass through the tissue, thus indicating that the plane is suitable for an ultrasound-guided puncture approach. Therefore, the ventral complex plays an important localization role in lumbar ultrasound. We hypothesized that CNNs might assist in the detection and segmentation of facet joints in ultrasound images. In this study, we proposed a deep learning method for recognizing ultrasound images of facet joints based on enhanced data annotation and CNNs. In the enhanced data annotation, the facet joint was considered as the first target and the posterior vertebral body as the second or auxiliary target.

## 2. Materials and Methods

Figure 1 shows the proposed deep learning-based ultrasonic image detection and segmentation method for facet joints. The facet joints were labeled as the primary target and the ventral complex was used as the auxiliary target. Preprocessing and data augmentation were conducted for each input ultrasound image. The deep learning network, the Mask Region-based CNN (Mask R-CNN), refs. [18,19] was used, in which ResNet101 and the feature pyramid network (FPN) were used as the backbone. The FPN was used to extract image features. The region proposal network (RPN) was used to find the region of interest (ROI). The ROI Align was used to transform all the proposed ROIs generated in the RPN process into a feature map of the same size, which was then reshaped into a one-dimensional vector, so as to facilitate the subsequent generation of masks, coordinates, and classifications for facet joint detection and segmentation [20].

### 2.1. Ultrasound Image Data

#### 2.1.1. Data Collection

In this study, the clinical data were collected from 300 patients. Most of these patients suffered from lower back pain and were undergoing pain treatment at the Department of Anesthesiology of Peking Union Medical College Hospital (PUMCH), Chinese Academy of Medical Sciences. The study protocol was numbered K22C2241 and approved by the Ethics Review Committee of PUMCH. All the patients received ultrasound scanning before and after pain treatment. The ultrasound images were captured by one or two expert sonographers using a SonoSite X-Porte scanner (Fujifilm, Tokyo, Japan), with a 2–5 MHz curved-array transducer (C60xp/5-2), a scanning depth of 7 cm, and a gain of 50%. The pixels of the original images in this study were 960 × 720. The collected data had the same scanning angle and a similar scanning field (Figure 2).

#### 2.1.2. Enhanced Data Annotation

Two professional anesthesiologists confirmed the inclusion of a facet joint and ventral complex in all the image data and processed the ultrasound image data to remove the patient’s name, physician’s unit, and physician’s name. Under the review and proofreading of the senior anesthesiologist, the annotator marked the outline of the facet joint and the ventral complex in each image. The marking software used in this study was LabelMe (version 4.5.13), an open annotation tool. In this study, we proposed an enhanced data annotation method for the facet joint using the facet joint as the first target and the ventral complex as the second target (Figure 3). In order to evaluate the influence of data annotation on deep learning, we considered two labeling manners: (i) a local labeling method (Figure 3b), which only involved facet joints and ventral association; (ii) a full labeling method (Figure 3a), which involved transverse processes, facet joints and even bone structures in the median line and ventral association.

#### 2.1.3. Data Cleaning

The process of data cleaning was to screen out abnormal data, including (1) images and marked data that were damaged; (2) data naming formats that did not meet the requirements; (3) incorrect marks; (4) incorrect association of the original image with the marked image; (5) images that did not contain a target object. In this study, a total of 391 original ultrasonic images remained after the data cleaning.

#### 2.1.4. Data Classification

The purpose of data classification was to divide the data into training sets and testing sets for the deep learning model. The 391 original ultrasonic images were divided into a training set composed of 356 images and a testing set composed of 35 images. Note that all images from the same patient were assigned to the same subset (training or testing).

### 2.2. Image Preprocessing and Data Augmentation

Image preprocessing was conducted to the input image, including image scaling and data augmentation. Since the input of the CNN must be images with the same width and height, one of the preprocessing tasks was to scale the original image to 256 × 256 pixels. In addition, the data augmentation method of horizontal flipping was used in this study according to medical knowledge. Generally, data augmentation should conform to certain physical meaning. Otherwise, inappropriate data augmentation may reduce the recognition accuracy. Specifically, each image in the training set was flipped left to right, so the number of images in the training set were doubled.

### 2.3. The Mask R-CNN

In this study, the ventral complex (denoted “E”) and the facet joint (denoted “F”) needed to be segmented in ultrasonic images. In theory, semantic segmentation and instance segmentation are both acceptable. Considering that, in some cases, E and F could be multiple and overlap, instance segmentation was adopted in this work. Mask R-CNN [18] is the most typical instance segmentation algorithm, in which target segmentation is achieved on the basis of target detection. Compared with U-Net [21] and its improved algorithms that can only achieve semantic segmentation, Mask R-CNN is more suitable for instance segmentation where the number of targets is small and the target occupies a low proportion of image pixels. This is because when the number of targets is small and the proportion of pixels is low, U-Net takes the background as an independent segmentation target and the number of network layers is low. In the training process, U-Net often quickly converged on a local optimality, ignoring all the targets and identifying the whole image as the background. In addition, the objective of this study was to detect and segment facet joints at the same time, for which Mask R-CNN was well-suited, while U-Net was only suitable for image segmentation. For these reasons, we considered that Mask R-CNN would be more suitable for the facet joint detection and segmentation task in this work.

#### 2.3.1. The FPN

The FPN is the backbone network of the Mask R-CNN [18], which is mainly used to extract image features (Figure 4). The FPN was divided into five layers in order to extract image features. Low-level features usually contain more details, such as textures and edges, but they may also contain a lot of noise. High-level features generally contain more semantic information (shape, position, etc.), but the spatial resolution is small, and the information loss is severe. Suppose the size of the original image is *H* × *W* pixels. The FPN extracted 5 features of different levels from low to high, whose feature sizes were *H*/2 × *W*/2, *H*/4 × *W*/4, *H*/8 × *W*/8, *H*/16 × *W*/16, and *H*/32 × *W*/32, respectively. In this study, *H* = 256 and *W* = 256. Standard ResNet [22] networks are usually used along with FPNs, such as the ResNet50 and ResNet101 networks. In this paper, the ResNet101 network was used.

#### 2.3.2. The RPN

The RPN found ROIs based on image features extracted from the FPN. The RPN can be understood as fulfilling two tasks. One is a classification task, and the other is a regression task. The regression task is to obtain the coordinates of the candidate box (the coordinates of the upper left and lower right of the candidate box), and the classification task is to determine whether there is a target in the candidate box (the probability of having a target). After the two tasks were completed, each candidate box with a probability score greater than 0.7 for an object was retained as a proposal.

#### 2.3.3. The ROI Align

The ROI Align was mainly used to transform all the proposed ROIs generated in the RPN process into feature maps of the same size. The feature maps are then reshaped into a one-dimensional vector. The size of the vector was 49 (i.e., 7 × 7) for facet joint detection and 196 (i.e., 14 × 14) for facet joint segmentation. Finally, the Mask R-CNN [15] produced segmentation results (mask), recognition results (coordinates), and categories of images through three independent fully connected networks (Figure 1).

### 2.4. Evaluation Metrics

During the testing phase, the results of the deep learning model were compared with the labeled data. The performance of the model was evaluated using the following metrics.

(1)True Positive (*TP*) represents the correct prediction of positive data.(2)True Negative (*TN*) represents the correct prediction of negative data.(3)False Positive (*FP*) represents the incorrect prediction of positive data.(4)False Negative (*FN*) represents the incorrect estimate of negative data.

The Intersection over Union (IoU) score is a standard performance measure for object segmentation tasks. Given a set of images, the IoU metric gives the similarity between the predicted region and the ground truth region of the objects presented in the set of images, and is defined by
(1)$$\mathrm{IoU}=\frac{TP}{FP+TP+FN}\;\mathrm{or}\;\mathrm{IoU}=\frac{A\cap B}{A\cup B}$$
where *A* is the area of model-based segmentation and *B* is the area of the ground truth.

With the deep learning algorithms, the success of the model depends on the result of the confusion matrix. The success rate of each algorithm for detecting facet joints was determined. The average precision (AP) was used to evaluate the facet joint detection and segmentation performance of the proposed method. The AP values of all categories were averaged to produce mAP as well. Specifically, AP50 and AP@50:5:95 were used. AP50 is defined as the AP when the IoU equals 50%. AP@50:5:95 is defined as the mean value of those APs corresponding to IoUs from 50% to 95% with a step of 5%.

### 2.5. Experimental Setup

The training platform for the experiments was a self-built server, based on a single 32G V100 PCIe GPU. The number of training iterations was set at 25,000, and completing all mini-batch trainings took approximately 8 min 50 s multiplied by 25,000 and divided by 20 (batch size = 16). That was approximately 219,166 s or about 61 h. The initial learning rate was set at 0.0001. The loss function of the Mask R-CNN was a multi-task loss *L*, *L* = *L*_C_ + *L*_D_ + *L*_S_, where the subscripts ‘C’, ‘D’, and ‘S’ denote classification, detection, and segmentation, respectively [18]. *L*_S_ is the average binary cross-entropy loss [18]. The testing sets were input to the trained deep learning model. A total of 421 ultrasound images were included in the dataset, and 420 valid data were obtained after data cleaning. Among them, 29 images did not contain any objects, so were deemed to constitute a negative set, and the remaining 391 image data were divided into a training set composed of 356 images and a validation set composed of 35 images. Validation sets and training sets were annotated in coco format.

### 2.6. Statistical Analysis

Categorical variables were expressed as frequencies and percentages, which were compared using *t*-tests. Statistical analyses were conducted using SPSS 20.0 (SPSS Inc., Chicago, IL, USA)

## 3. Results

In Figure 5, the green area is the inspected and segmented facet joint structure, and the orange area is the inspected and segmented ventral complex. The figure shows that the results for the facet joint structures inspected and segmented using the proposed method are close to the results of manual annotations by human experts, indicating the good performance of the enhanced data annotation method and the deep learning model adopted in this work.

Table 1 shows the effects of data augmentation on the AP of facet joint detection and segmentation using the proposed method. Compared to using no data augmentation, using horizontal flipping for data augmentation improves the AP50 and AP@50:5:95 values in both detection and segmentation. Based on the symmetry of the facet joints, we considered that horizontal flipping is an effective method for data augmentation.

Table 2 shows the effects of data annotation methods on the AP of facet joint detection and segmentation using the proposed method. Although it is generally believed that using the full labeling method reduces the false detection rate and improves the recognition accuracy of the target, our results show that, for the task of facet joint detection and segmentation in ultrasound images, using the local labeling method produces higher AP50 and AP@50:5:95 values in facet joint segmentation using the proposed method.

## 4. Discussion

The application of AI in ultrasound imaging is currently a hot topic, especially in the fields of liver, cardiovascular, thyroid, and musculoskeletal systems [23,24,25,26]. AI techniques include conventional machine learning methods and deep learning methods. CNNs are types of deep learning techniques, biologically inspired neural networks that mimic the physiology of the visual cortex by responding differently to specific features [27]. CNNs are composed of a series of convolutional layers, followed by a pooling layer, and finally a fully connected layer. CNNs have been applied to ultrasound images to identify and recognize different target objects, such as neural vascular structures [28] (12), left ventricles [29,30], breast tumors, and the spine [31]. However, CNNs and other deep learning techniques have not been applied to ultrasound image analysis for facet joint detection and segmentation.

In recent years, musculoskeletal ultrasound has been widely used in the field of rehabilitation, anesthesiology, orthopedics, and other fields for puncture positioning and real-time guidance. It is well known that the bony structures of the lumbar spine, including spinous processes, vertebral arch plates, facet joints and transverse processes, have typical ultrasound characteristics. Ultrasound scans of the lumbar spine structure can be applied to minimally invasive treatments such as spinal or epidural anesthesia, lumbar nerve blocks, quadratus lumborum plane blocks, and endoscopic foraminal surgery. Among them, the recognition of the bone structure of the facet joint is particularly of interest. On one hand, the posterior medial branch of the lumbar nerve is close to the rear of the facet joint. On the other hand, the blocking point of the posterior medial branch is the recess between the outer side of the upper facet and the proximal edge of the transverse process. The structure of the facet joint can be accurately identified using the characteristic ultrasound images of the upper vertebral arch plate, the lower facet, the facet joint, and the transverse process. Real-time ultrasound guidance can accurately locate the position of the puncture needle and avoid intravascular injection.

In the early stage of this study, we supposed that the bone structure of the facet joint would be connected to the transverse process and spinous process, so we labeled all the relevant bone structures in the ultrasound images, i.e., using the full labeling method. But soon we found that this full labeling approach resulted in poor model accuracy. The main reason may be that the range of bone structures is large in full labeling, and the boundary is not so clear, resulting in a large boundary error in deep learning. After recognizing this, we chose a local labeling method, that is, only the bony structure of the facet joint area was labeled. With this method, the boundary can be relatively clear, and the area that needs to be labeled is relatively small. It turns out that the local labeling method has better accuracy. Although we generally believed that labeling more features would reduce the false detection rate of the target and improve the recognition accuracy of the target, our experimental results showed that a larger scope of labeling is not better. We should meet the requirements of medical scenarios and computer vision algorithms.

In this study, we investigated an approach using CNNs and enhanced data annotation methods for facet joint detection and segmentation in ultrasound images. Specifically, the Mask R-CNN and enhanced annotation of the facet joint and ventral complex yields satisfying detection and segmentation performance. To the best of our knowledge, this work is the first to demonstrate the feasibility of deep learning models in detecting and segmenting facet joints in ultrasound images.

The considerations of the enhanced data annotation method proposed in this work are discussed. The ventral complex can assist in the detection and recognition of the facet joint. For instance, if there are facet joint–transverse process objects appearing in ultrasound images, with no dura mater appearing, then the facet joint–transverse process object detection may be true, or false. However, if there are facet joint–transverse process objects appearing in ultrasound images, with dura mater appearing, then facet joint–transverse process object detection should be true. The dura mater is the structure inside the spinal canal. If the dura mater appears in the ultrasound image, it means that the ultrasonic scanning plane must be at the level of the intervertebral space, and the facet joint–transverse process is also at the level of the intervertebral space. Therefore, the facet joint–transverse process must be true at the level where the dura mater can appear. However, if the facet joint–transverse process appears in the absence of dura mater, it may be that the ossification and calcification of intervertebral tissue has caused a failure to recognize the dura mater in ultrasound images, or it may be due to other tissues whose structures are similar to the facet joint–transverse process. With the enhanced data annotation method and combined identification of the facet joint and ventral complex, the negative sets can be correctly recognized (Figure 6).

It should be noted that facet joint detection and segmentation in ultrasonic images is not the final goal. Clinicians are concerned about the accuracy of the puncture of each target. Based on the segmentation results from our deep learning model, the search inflection point of the segmentation (binary images) can be computed. As the final detection result of the target, the positioning error of the facet joint–transverse process can be controlled within an available range (typically 5 mm). This can effectively improve the accuracy of each puncture. Specifically, after the target detection and segmentation results are obtained, the search inflection points of the recognition coordinates can be obtained according to the recognition coordinates of the lumbar facet joint–transverse process and the dura mater, and the search inflection point can be added to the recognition coordinates to obtain the target recognition coordinates.

Finally, this study has some limitations. First, the size of the ultrasound image samples was relatively limited. It is worth noting that after horizontal flipping was performed on the training set, the detection AP50 of the test set increased from 93.8% to 94.2%, and the segmentation AP50 of the test set increased from 66.2% to 69.9%. Our experiments showed that the accuracy of the model can be increased by enlarging the amount of training sets in different ways. That is to say, the size of training sets should also be increased to get better experimental results. In addition, the full labeling method performs better for detection, and the local labeling method for segmentation (Table 2), so combining the respective neural network layers/branches for each model may be considered in future work. Last but not least, the images were collected at a single center with a single scanner. In future work, more images collected at different centers with different scanners may be used to further validate and improve the performance of deep learning models in facet joint detection and segmentation in ultrasound images.

## 5. Conclusions

In conclusion, this study is the first to present a deep learning method for facet joint detection and segmentation in ultrasound images based on enhanced data annotation and the Mask R-CNN. The feasibility and potential of deep learning techniques in facet joint ultrasound image analysis have been demonstrated. In the future, the proposed method may be used in the field of pain management and medical education.

## Figures and Tables

**Figure 1 diagnostics-14-00755-f001:**
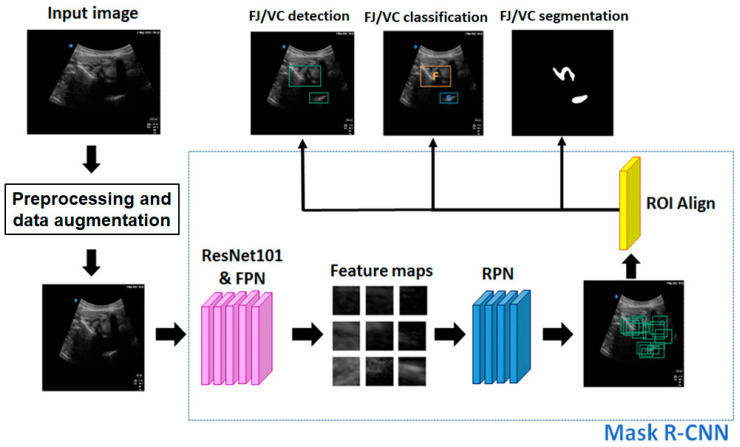
Flow chart of facet joint (FJ) and ventral complex (VC) detection and segmentation using the proposed method. In FJ/VC classification, the orange box represents the detected FJ (denoted “F”) and the blue box indicates the detected VC (denoted “E”). FPN = feature pyramid network; RPN = region proposal network; ROI = region of interest.

**Figure 2 diagnostics-14-00755-f002:**
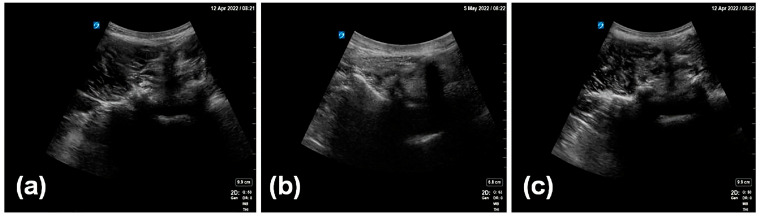
Examples of collected ultrasound images (**a**–**c**).

**Figure 3 diagnostics-14-00755-f003:**
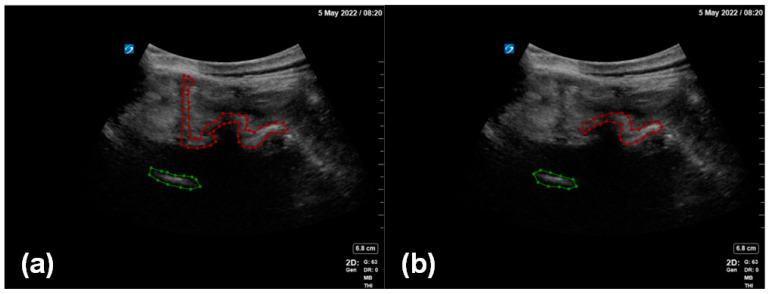
Two methods of enhanced data annotation: (**a**) full labeling method; (**b**) local labeling method. The red annotation indicates the facet joint, and the green annotation indicates the ventral complex.

**Figure 4 diagnostics-14-00755-f004:**
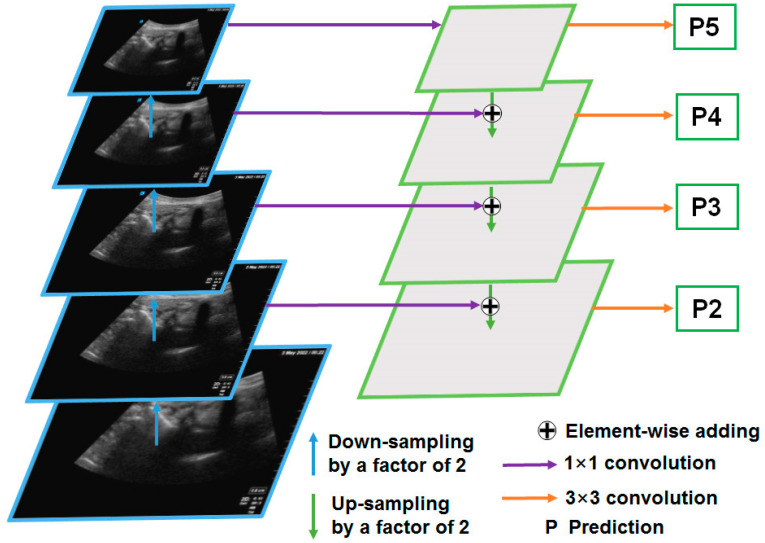
Structure of the feature pyramid network.

**Figure 5 diagnostics-14-00755-f005:**
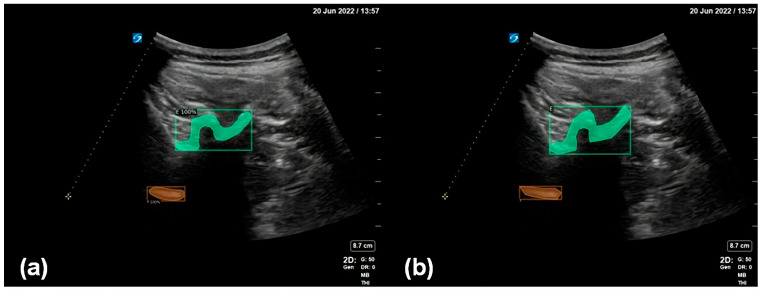
(**a**) Detection and segmentation of the facet joint (green) and the ventral complex (orange) using the proposed method. (**b**) The facet joint (green) and the ventral complex (orange) as manually annotated by human experts.

**Figure 6 diagnostics-14-00755-f006:**
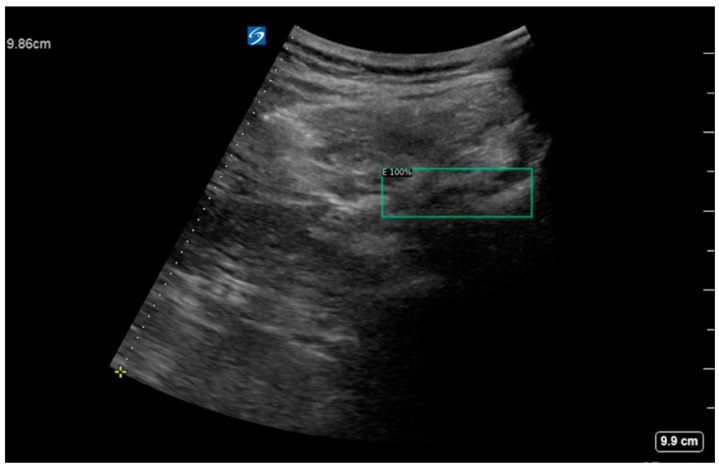
Detection of a negative set using the proposed method.

**Table 1 diagnostics-14-00755-t001:** Effects of data augmentation on the average precision (AP) of facet joint detection and segmentation using the proposed method.

Data Augmentation	Detection		Segmentation	
	AP50	AP@50:5:95	AP50	AP@50:5:95
None	92.17%	54.73%	82.86%	36.71%
Horizontal flip	92.88%	54.95%	90.01%	39.86%
*p*	<0.001	<0.001	<0.001	<0.001

**Table 2 diagnostics-14-00755-t002:** Effects of data annotation methods on the average precision (AP) of facet joint detection and segmentation using the proposed method.

Data Annotation Area	Detection		Segmentation	
	AP50	AP@50:5:95	AP50	AP@50:5:95
Full labeling method	92.17%	54.73%	82.86%	36.71%
Local labeling method	98.57%	63.86%	90.75%	44.01%
*p*	<0.001	<0.001	<0.001	<0.001

## Data Availability

The datasets generated and analyzed and the code during the current study are available from the corresponding author on reasonable request.

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
