# Peer review of "Deep Learning Detection and Segmentation of Facet Joints in Ultrasound Images Based on Convolutional Neural Networks and Enhanced Data Annotation"

_diagnostics, 2024, doi:10.3390/diagnostics14070755_

Round 1

Reviewer 1 Report

Comments and Suggestions for Authors

In this paper, the authors tackled an interesting problem of automated detection and segmentation of facet joints in ultrasound images. The topic is certainly worthy of investigation and easily falls into the scope of the journal. There are, however, quite a number of issues which should be thoroughly addressed before the manuscript could be considered for publication:

1.      The authors should make sure that each abbreviation is defined once at its first use (see e.g., CNN in the abstract).

2.      The manuscript would greatly benefit from careful proofreading – there are unclear and vague statements around the manuscript (also in the abstract, e.g., “The training set number, data augmentation and data annotation methods could improve the average precision” – the authors probably mean the size of the training set).

3.      Is it necessary to have three digits after the decimal point in the abstract? (90.350% and 84.966%).

4.      It would be useful to announce the structure of the paper in the introductory section.

5.      The authors should make sure that it is possible to reproduce the entire processing pipeline based on the description provided. In its current form, it would be challenging (if possible at all) to do that – as an example, it is unclear if data augmentation was executed over the training data only? Please provide precise hyperparameters for data augmentation and all other steps in the processing chain.

6.      Please provide clear and convincing motivation behind selecting specific models and algorithms in the processing chain, especially given that there are quite a number of possibilities (e.g., in the context of deep learning architectures).

7.      Please avoid section headers containing abbreviations only, e.g., RPN, FPN and so forth.

8.      Why the size of the test set is different in Table 1 for two sizes of the training sets? If the test sets are different then these metrics cannot be directly compared.

9.      In a similar vein to one of my previous comments – we are currently facing the reproducibility crisis in the machine learning research. To this end, please provide a link to the repository containing the implementation of the pipeline, together with the datasets used in this study.

10.   The authors should compare their technique with other methods from the literature, such as fully-convolutional networks, U-Nets and preferably nnU-Nets, which established the state of the art in the field of medical image segmentation.

11.   Please back up the experimental study with appropriate statistical testing – are the differences between the investigated methods statistically significant? Please report appropriate p-values.

Comments on the Quality of English Language

The manuscript should be carefully proofread.

Author Response

Thank you very much for taking the time to review this manuscript. Please see the attachment

Reviewer 2 Report

Comments and Suggestions for Authors

This manuscript describes the development and validation of a Mask R-CNN model, for detecting and segmenting the facet joint (and ventral complex) from ultrasound images. After data clearning and elimination of images without annotated objects, 356 images were assigned to the training set, and the remaining 35 images to the validation set. Results for facet joint detection and segmentation varying training set size, data augmentation and full/local labeling annotations were reported.

Some issues might be considered:

1. In Line 146 (and 226), it is stated that 420 images were obtained after data cleaning, with 29 images then rejected for not containing any object. It might be clarified as to whether this additional rejection should be included as part of the original data cleaning, and what was the single image rejected after data cleaning, rejected for.

2. Moreover, since the 391 images were obtained from 300 patients (Line 113), it appears that some patients would have more than one image. As such, it might be clarified as to whether all images from the same patient were assigned to the same subset (training or validation).

3. In Line 188, it is stated that "After the two tasks are completed, the candidate box with a probability score greater than 0.7 for an object is retained as a proposal". Given this, it might be clarified as to what happens when overlapping candidate boxes (e.g. in Figure 1) have probability scores greater than 0.7, since each candidate box has its associated object feature map/mask. Are the overlapping masks also combined?

4. In Line 192, it is stated that ROI Align transforms all proposed ROIs into feature maps of the same size, which are then reshaped into a one-dimensional vector. The size of the vector might be stated.

5. It is then stated that Mask R-CNN obtains segmentation results, recognition results and categories of images through three independent fully-connected networks. Figure 1 only shows two final branches - this might be clarified. Also, the specific model architecture (including layers) & hyperparameters, and losses corresponding to each network/output branch, might be clarified.

6. In Line 210, the definition of IoU is given. The IoU metric might also be reported for each of the result tables, at the chosen threshold.

7. In Line 213, "thses" might be "these".

8. In Table 1, it might be clarified if the size-284 training set is a subset of the size-356 training set.

9. In Table 3, the full labeling method performs better for detection, and the local labeling method for segmentation. As such, would it make sense to combine the respective neural network layers/branches, for each model?

10. While it is claimed that Mask R-CNN outperforms U-Net on the task, it would be best justified with empirical results using U-Net and other comparison methods, if possible.

Comments on the Quality of English Language

N/A

Author Response

(The authors gave the same response as above.)

Round 2

Reviewer 1 Report

Comments and Suggestions for Authors

Thank you for addressing most of my concerns. I would, however, still encourage the authors to do the full proofreading to improve the language and presentation.

Comments on the Quality of English Language

The manuscript would still benefit from proofreading, perhaps with a help of a native speaking colleague.

Author Response

Thank you for your suggestion, the language and presentation has been improved. I hope the manuscript could meet your requirement.

Reviewer 2 Report

Comments and Suggestions for Authors

We thank the authors for addressing our previous comments. In Figure 1, "Prepressing" might be "Preprocessing".

Comments on the Quality of English Language

N/A

Author Response

Thank you for your suggestion. The language and presentation has been improved.  Fig 1 has been corrected. I hope the manuscript could meet your requirement.